# Research on the coordination of fiscal structure and high-quality economic development: An empirical analysis with the example of Anhui province in China

Yunlei Zhou[1], Xiaoyu He[1], Shengsheng Li[2¤]*

1 School of Finance and Public Administration, Anhui University of Finance and Economics, Bengbu, China,
2 School of Economics, Anhui University, Hefei, China

¤ Current address: School of Business, Fuyang Normal University, Fuyang, China
* lisheng2@foxmail.com

**Data Availability Statement:** https://gitee.com/MoRanHuiShou2017/plos-one/tree/master/.

**Funding:** For this work, the first author(Yunlei Zhou) was supported by the 2021 Research

## Abstract

Based on the panel data of 16 cities in Anhui from 2010–2018, this paper measures the index system of fiscal structure and high-quality economic development in Anhui using the entropy weight method, and empirically analyzes the coordinated development level of fiscal structure and high-quality economic development in Anhui using coupled coordination degree model. The study finds that: (1) the overall structure of Anhui's fiscal expenditure is characterized by "service-oriented and investment-oriented", and there is a phenomenon that contradicts "Wagner Principle", and there are also spatial and temporal differences in Anhui's tax structure. (2) The level of high-quality development of Anhui economy shows a steady upward trend, but is still at a low level. (3) The level of coordinated development of fiscal structure and high-quality economic development is still low, and the overall situation is "on the verge of disorder" or "barely coordinated". (4) Regionally, the overall coordination of fiscal expenditure structure, tax structure, and high-quality economic development in southern Anhui shows a decreasing trend, while the overall coordination in central and northern Anhui shows an increasing trend, so that southern Anhui has been or will be surpassed by northern and middle Anhui, and the growth rate of middle Anhui is faster than that of northern Anhui.

## 1. Introduction

China's economy has shifted from the stage of high-speed growth to the stage of high-quality development [1]. Promoting high-quality economic development is an inevitable choice to adapt to the current new normal of economic development, and is the embodiment of the five new development concepts of innovation, coordination, green, openness, and sharing [2], and is also an inevitable requirement to adapt to the changes in the main contradictions of Chinese society, follow the laws of economic development and maintain healthy economic development. From Adam Smith's "small government" to Marshall's partial equilibrium analysis,

Projects on Humanities and Social Sciences in Anhui Universities under grant SK2021A0229, the Project of Anhui Ecological and Economic Development Research Center of China under grant AHST2019016, AHST2021010, and AHST2019011, and the Social Science Planning in Anhui Province of China under grant AHSKZ2019D018. The second author(Xiaoyu He) was supported the by the National Social Science Youth Fund of China under grant 19CJY055. The third author(Shengsheng Li) was supported by the Project of Anhui Ecological and Economic Development Research Center of China under grant AHST2021007. The funders had no role in study design, data collection and analysis, decision to publish, or preparation of the manuscript.

**Competing interests:** The authors have declared that no competing interests exist.

Vallas' general equilibrium analysis, the theory of perfect competition in welfare economics, to the "big government" advocated by the neoclassical school of synthesis after Keynes, to monetarism and the rational expectations school, all of them have reconceptualized the limits of the role of government and the scope of government activities. The aforementioned scholars have continued to explore and deepen their understanding of market resource allocation and government fiscal intervention. Finance is the foundation and important pillar of national governance [3], and it still plays a crucial role in the current process of high-quality economic development.

Anhui Province is located in East China, the Yangtze River Delta hinterland, spans the Yangtze River, and the Huai River north and south, with the Huabei Plain, the Jianghuai Hills, the mountains of southern Anhui three natural areas, the geographical position is superior, rich in products. In the context of the Yangtze River Delta integration strategy, promoting high-quality economic development in the Yangtze River Delta region is one of the key tasks in the implementation of the Yangtze River Delta integration strategy [4]. As one of the three provinces and one city member of the Yangtze River Delta, Anhui Province ushered in a period of opportunity for rapid development. And what is the comprehensive development level of the two systems of fiscal structure and such as high-quality development of Anhui economy in the new era? What is the relationship between them? What is the degree of coordination between the two systems of fiscal structure and such as the high-quality development of Anhui economy?

Coupled and coordinated development means that two or more subsystems interact with each other to produce a synergistic amplification effect [5]. Based on China's five development concepts, this paper analyzes the coupled and coordinated development mechanism of fiscal structure and high-quality economic development from five aspects: innovative economic development, coordinated economic development, green economic development, open economic development, and shared economic development, which is shown in Fig 1.

1. Fiscal structure and innovative economic development
   Innovation is the driving force of high-quality economic development, and it is crucial to promote innovative economic development [6]. Faced with the dilemma of the capital constraint of enterprise innovation and the lack of incentive to innovate, the government often uses macroeconomic policies such as fiscal policy as a grip to stimulate enterprise innovation. Investment expenditures such as general public service expenditures, transportation expenditures, and government debt service expenditures are conducive to infrastructure construction and the introduction of advanced technology and equipment, providing an excellent hardware environment for economic innovation and development, while social expenditures such as education expenditures, science and technology expenditures and medical health and family planning expenditures directly stimulate various enterprises to innovate, realize "promoting revenue with expenditures" and cultivate "new financial resources", thus promoting the rational and effective allocation of financial resources and realizing a virtuous cycle of development. Taxation plays a regulatory role in the process of economic development that cannot be ignored [7]. In terms of fiscal structure, the literature [8] emphasizes the need to revise the approach to monetary and fiscal policy, placing greater emphasis on the coordination and harmonization of macroeconomic regulatory instruments to ensure sustainable economic growth in the long term. The literature [9] combined financial innovation and economic growth to construct a corresponding synergy model, and discovered the synergistic development relationship by studying the degree of synergy in the past period, and found that the coordinated development of financial innovation and economic growth would produce an overall synergistic effect of "1+1>2".

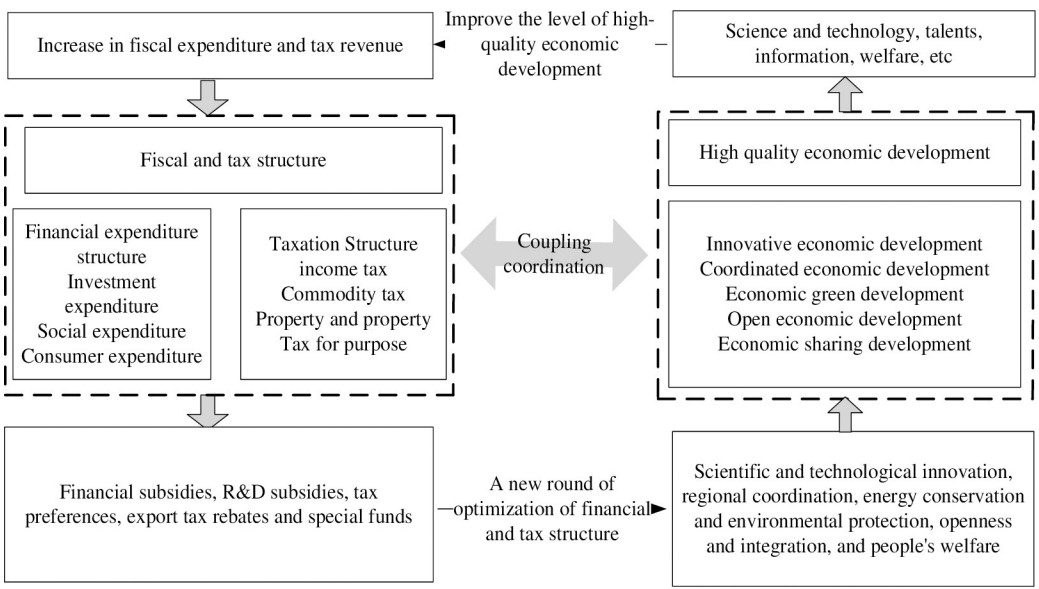

**Fig 1. Mechanism of coupling and coordination between financial and tax structure and high-quality economic development.**

Therefore, the improvement in macro tax burden is conducive to higher government fiscal spending, which in turn improves economic development [10]. For example, the tax concessions for corporate income tax for high-tech enterprises and the full implementation of the policy of business tax to value-added tax. Tax incentives for property tax, urban land use tax, and other property and behavioral taxes are conducive to reducing the tax burden of enterprises, improving their profitability, increasing their enthusiasm for R&D investment, and thus promoting innovative economic development.

2. Fiscal structure and coordinated economic development
   As an important tool for the government to regulate the economy, fiscal expenditure instruments and taxation instruments play an important role in stimulating coordinated economic development. The results of previous studies in the literature found that the degree of coupling and coordination of government support, financial support, and innovation had a significant positive impact on economic development [11]. The government can work on the industrial structure through government procurement and government investment in fiscal spending to achieve the goals of industrial policy, and for industries that are to be developed in a key way as well as those related to high-quality development, the government promotes the development of the industry by means of financial subsidies and subsidized loans. Fiscal expenditures on education, health care, and family planning can promote the increase of residents' consumption level [12], while taxation mainly influences consumers' choices through its income effect and substitution effect, especially the personal income tax concessions in taxation have an income effect on residents and their disposable income increases, thus contributing to the increase of residents' consumption level. The increase of guaranteed financial expenditure can narrow the income gap between urban and rural areas and regulate the dual structure of urban and rural areas, and the implementation of the over-progressive tax rate of personal income tax is also conducive to narrowing the gap between the rich and the poor and promoting coordinated economic development [13].

3. Fiscal structure and green economic development
   Fiscal and taxation policies act on green economic development through positive and negative incentives, mainly in the following aspects: First, fiscal expenditure policies such as government green procurement, investment in pollution control, and tax incentives for enterprises that meet energy conservation and environmental protection standards can motivate enterprises to save energy and reduce emissions and green production. Second, the fiscal expenditure on science and technology, research, and development subsidies can stimulate enterprise innovation, promote the transformation and upgrading of enterprises, improve the efficiency of resource and energy utilization, and promote the green development of enterprises. Thirdly, the environmental taxes such as car and boat tax and resource tax can play a restraining role in the production of enterprises [14] and promote green development through negative incentives such as "punishment". Fiscal R&D expenditures and education expenditures promote green economic growth through technological and human capital-intensive activities, respectively [15]. There is a close relationship between fiscal expenditure structure and green economic growth, and a reasonable and appropriate selection and allocation of fiscal expenditure structure is conducive to promoting the growth of China's overall green economy [16].

4. Fiscal structure and open economic development
   The impact of fiscal policy on the open development of the economy is mainly reflected in the exchange rate and prices [17]. On the one hand, an active fiscal policy of increasing fiscal spending and reducing taxes will stimulate aggregate social demand and promote economic development, while bringing about an increase in the national exchange rate and higher commodity prices, which will have a crowding-out effect on commodity exports. On the other hand, through policies such as fiscal subsidies and tax preferences for import and export commodities and their downstream product elements, thereby influence the relative prices of imported and exported commodities, which in turn affects the level of trade openness [18]. In addition, when enterprises face an increase in tax burden such as income tax, commodity tax, and property tax, it will also force enterprises to develop foreign markets and improve their international competitiveness.

5. Fiscal structure and shared economic development

   Fiscal expenditures on education, health care, and family planning are conducive to improving the level of school teachers and residents' education, improving the level of residents' medical coverage, and raising the index of residents' welfare and happiness [19]. Tax policies influence residents' choices through their income effects and substitution effects [20]. On the one hand, the collection of personal income tax will lower residents' disposable income and reduce their consumption level, thus lowering their welfare level; on the other hand, the preferential policies on property tax, urban land use tax, and value-added tax for non-profit institutions such as schools and hospitals are conducive to guaranteeing residents' basic public service level and promoting shared economic development.

   In summary, the government through the optimization of the fiscal structure of the high quality of economic development to produce a positive promotion effect, bringing science and technology, talent, information, and other factors to increase, which further promotes the level of high-quality economic development, fiscal spending scale and taxation scale increase, which in turn requires a new round of fiscal structure optimization, so present a spiral of virtuous cycle of development trend.

   In the previous literature, no lack of literature studies the coupling relationship between the two systems. For example, literature [21] studied the coupling coordination level between green economy and green finance, and literature [22] analyzed the coupling coordination relationship between green finance and economic growth. The literature [23] found that the level

of coupling coordination between green finance and ecological environment in the Yangtze River basin developed from the uncoordinated stage to the well-coordinated stage. However, there is a lack of literature on the coupling and coordination between financial structure and economic high quality, and literature similar to our topic mainly includes literature [24] which studied the coupling and coordination between green finance and economic high-quality development. Literature examining the coupled coordination between the two systems with the theme of China also includes literature [25, 26]. Among the evaluation models of coordination, the coupled coordination model has proven to be an effective analysis method [27].

Given this background, this paper selects the panel data of 16 cities in Anhui from 2010–2018 as samples, constructs the index system of fiscal structure and high-quality development of Anhui economy, theoretically analyzes the mechanism of the coordinated development of fiscal structure and high-quality development of Anhui economy, and adopts the coordination degree model to measure the coordination degree of two systems of fiscal structure and high-quality development of Anhui economy, and conducts an in-depth analysis of the spatial and temporal change characteristics of their coordination degree, which is of great significance to promote the high-quality development of Anhui economy and help the high-quality development of Yangtze River Delta regional economy.

## 2. Methodology

### 2.1. Entropy weight method

The entropy weighting method, as an objective assignment method, avoids to a certain extent the error nature of subjectively assigned weights [28], and its calculation process is as follows.

(1) Form the original data matrix.

The existing evaluated object $M = (M_1, M_2, M_3, \cdots, M_m)$, evaluation indicator $D = (D_1, D_2, D_3, \cdots, D_n)$, and the value of the evaluated object $M_i$ to the indicator $D_j$ is recorded as C, then the original data matrix formed is

$$X = \begin{bmatrix} x_{11} & x_{12} & \cdots & x_{1n} \\ x_{21} & x_{22} & \cdots & x_{2n} \\ \cdots & \cdots & \cdots & \cdots \\ x_{m1} & x_{m2} & \cdots & x_{mn} \end{bmatrix}_{m \times n} \tag{1}$$

where $x_{ij}$ is the corresponding indicator value of the i-th city under the j-th indicator.

(2) Dimensionless processing of the original matrix.

When the selected indicator is a positive indicator, the dimensionless treatment is calculated as:

$$v_{ij} = \frac{x_{ij} - \min(x_j)}{\max(x_j) - \min(x_j)} \tag{2}$$

When the selected indicator is a negative indicator, the dimensionless treatment is calculated as:

$$v_{ij} = \frac{\max(x_j) - x_{ij}}{\max(x_j) - \min(x_j)} \tag{3}$$

The matrix after dimensionless processing of each indicator through Eqs (2) and (3) is

$$V_{ij} = \begin{bmatrix} v_{11} & v_{12} & \cdots & v_{1n} \\ v_{21} & v_{22} & \cdots & v_{2n} \\ \cdots & \cdots & \cdots & \cdots \\ v_{m1} & v_{m2} & \cdots & v_{mn} \end{bmatrix}_{m \times n} \tag{4}$$

(3) Calculate the characteristic weight of the ith city under the jth indicator.

Let the characteristic weight of the i-th city under the jth indicator be X. Then X is

$$p_{ij} = v_{ij} / \sum_{i=1}^{m} v_{ij} \tag{5}$$

Then, based on the value of $p_{ij}$, the entropy value $e_j$ of the j indicators can be calculated as

$$e_j = -1/\ln(m) \sum_{i=1}^{m} p_{ij} \cdot \ln(p_{ij}) \tag{6}$$

When the data is processed dimensionless, there will be the case of $p_{ij} = 0$. It is stipulated that when $p_{ij} = 0$, $\ln p_{ij} = 0$.

(4) Determine the entropy method weights.

After obtaining $e_j$ according to Eq (6), the coefficient of variation $d_j$ for the jth indicator can be calculated, and the coefficient of variation $d_j$ is $d_j = 1 - e_j$. The larger the value of $d_j$, the greater the amount of information provided by the indicator and the greater the weight should be given. Then the entropy method weight $w_j$ for this value is defined as

$$w_j = d_j / \sum_{j=1}^{n} d_j \tag{7}$$

(5) Calculate the entropy value.

The comprehensive evaluation level of fiscal expenditure structure (CEV) is obtained by using the dimensionless data weighted by the following formula.

$$CEV_i = \sum_{j=1}^{n} w_j p_{ij} \tag{8}$$

## 2.2. System coordination measurement method

The phenomenon of two or more systems interacting and influencing each other is known as coupling, and the degree of coordination can measure the magnitude of coupling. Therefore, this paper draws on the coupling coefficient model in physics and introduces the coupling coordination degree method to measure the degree of coordinated development of fiscal structure and high-quality economic development. The specific calculation formula is:

$$C = \left\{ X \times Y / \left( \frac{X+Y}{2} \right)^2 \right\}^k$$
$$I = \theta X + \delta Y \tag{9}$$
$$D = \sqrt{C \times I}$$

where $C$ is the coupling degree; $D$ is the coordination degree; $X$ is the comprehensive

**Table 1. The coordination judgment criteria.**

| Coherence value | System coordination level | Coherence value | System coordination level |
|---|---|---|---|
| 0.00–0.09 | extreme disorder | 0.50–0.59 | barely coordination |
| 0.10–0.19 | severe disorder | 0.60–0.69 | primary coordination |
| 0.20–0.29 | moderate disorder | 0.70–0.79 | intermediate coordination |
| 0.30–0.39 | mild disorder | 0.80–0.89 | good coordination |
| 0.40–0.49 | nearly disorder | 0.90–1.00 | quality coordination |

evaluation level of fiscal expenditure structure and the comprehensive evaluation level of tax structure, and $Y$ is the level of high-quality economic development; $k$ is the adjustment coefficient (here it takes the value of 2); $I$ is the comprehensive development level of fiscal and tax structure and high-quality economic development. $\theta$ and $\delta$ are the index weights. Since the optimization of the tax structure can stimulate high-quality economic development, and the improvement of the high-quality economic development promotes the optimization of the tax structure, the two interact with each other and complement each other, so take $\theta = \delta = 0.5$. The value of the coordination degree $D$ ranges between 0 and 1, and its specific judgment criteria are shown in Table 1.

## 3. Indicators

### 3.1. Construction of financial and tax structure index system

(1) Fiscal expenditure structure.

This paper examines the structure of fiscal expenditures in terms of investment expenditures, social expenditures, and consumption expenditures based on the reform of the classification of fiscal revenues and expenditures of the Chinese government in 2007. After 2007 (inclusive), the Chinese government divided the structure of fiscal expenditures into three categories: investment expenditures, social expenditures and consumption expenditures according to three government functions: economic construction, social services and political management (Among them, investment expenditures include general public services, transportation, government debt service expenditures; social expenditures include education, science and technology, culture, sports and media, social security and employment, health care, environmental protection (energy conservation and protection), agriculture, forestry and water affairs, post-earthquake restoration and reconstruction expenditures, land, resources, meteorology and other affairs, housing security expenditures; consumer expenditures include foreign affairs, national defense, public security, financial supervision and other affairs, industrial, commercial and financial affairs, extractive power information and other affairs (resource exploration and power information and other affairs), commercial services and other affairs, grain and oil supplies reserve management affairs, other expenditures.).

Investment expenditures (IE) are the sum of general public service expenditures, transportation expenditures, and government debt service expenditures. Social expenditure (SE) is the sum of education expenditure, science and technology expenditure, culture, sports, and media expenditure, social security, and employment expenditure, medical health and family planning expenditure, energy conservation and environmental protection expenditure, urban and rural community expenditure, agriculture, forestry and water expenditure, land, sea and weather expenditure, and housing security expenditure. Consumption expenditure (CE) is the sum of national defense expenditure, public security expenditure, resource exploration and information expenditure, financial supervision, and other expenditure, commercial service expenditure, food, and oil supplies reserve expenditure, and other expenditure. All the above expenditures exclude

**Table 2. Fiscal structure evaluation index system.**

| Entropy value | indicator | indicator measurement method | indicator property |
|---|---|---|---|
| FESI | IE | IE share of GDP (%) | positive indicator |
| | SE | SE share of GDP (%) | positive indicator |
| | CE | CE share of GDP (%) | positive indicator |
| TSEI | INT | INT share of GDP (%) | positive indicator |
| | COT | COT share of GDP (%) | positive indicator |
| | PCT | PCT share of GDP (%) | positive indicator |

the influence of regional economic scale factors. Meanwhile, to analyze the coordination between fiscal expenditure structure and high-quality economic development, the entropy weight method is used to measure the comprehensive evaluation level of fiscal expenditure structure (FESI).

(2) Tax structure.

In this paper, we analyze the tax structure mainly in terms of income tax, commodity tax, property, and behavioral purpose tax with reference to the tax system classification of International Monetary Fund (IMF) and Organization for Economic Co-operation and Development (OECD). Given data availability, income tax (INT) is the sum of corporate income tax and personal income tax; commodity tax (COT) is the sum of value-added tax, business tax, resource tax, urban maintenance and construction tax and tobacco tax; property and conduct purpose tax (PCT) is the sum of property tax, stamp tax, urban land use tax, land value-added tax, vehicle, and vessel tax, arable land occupation tax and deed tax. All the above indicators exclude the influence of regional economic scale factors. In order to analyze the coordination degree of tax structure and high-quality development of Anhui economy, the comprehensive evaluation level of tax structure (TSEI) is measured by the entropy weight method. The specific fiscal structure index evaluation system is shown in Table 2.

## 3.2. Economic quality development index system

China's high-quality economic development is a development guided by the five major development concepts of innovation, coordination, green, openness, and sharing [29, 30], driving high-quality economic development with innovation, balancing high-quality economic development with coordination, brightening high-quality economic development with green, integrating high-quality economic development with openness and achieving high-quality economic development with sharing. This paper constructs the evaluation system of high-quality development indexes of Anhui economy with the five development concepts as the guide, as shown in Table 3.

After the index system is constructed, the entropy weighting method is used to objectively assign weights to the index system to calculate the level of high-quality economic development (HQED), as well as the level of innovative economic development (IND), coordinated economic development (COD), green economic development (GRD), open economic development (OPD) and shared economic development (SHD).

The indicators constructed by the indicator system are taken from the statistical yearbooks of each city in Anhui Province.

## 4. Results and discussion

### 4.1. The overall situation of Anhui's financial tax structure

Regarding the structure of fiscal expenditure, this paper mainly examines the profile of Anhui's fiscal expenditure structure in terms of investment expenditure (IE), social expenditure (SE),

**Table 3. Economic quality development evaluation index system.**

| | entropy value | indicator | sub-indicator | indicator property |
|---|---|---|---|---|
| High Quality Economic Development Level (HQED) | Innovative economic development (IND) | R&D investment ($x_1$) | R&D expenditure as a proportion of GDP (%) | positive indicator |
| | | | Number of R&D personnel | positive indicator |
| | | | Number of R&D institutions | positive indicator |
| | | talent development ($x_2$) | Number of students enrolled in general higher education institutions | positive indicator |
| | | scientific and technological achievements ($x_3$) | Number of patent applications granted | positive indicator |
| | | transformation of results ($x_4$) | GDP growth rate (%) | positive indicator |
| | | | Number of industrial enterprises above the scale | positive indicator |
| | | | Industrial added value of industrial enterprises above the scale (billion yuan) | positive indicator |
| | Coordinated Economic Development (COD) | industry structure ($x_5$) | Degree of industrial structure rationalization [31] | negative indicator |
| | | | Advanced degree of industrial structure [31] | positive indicator |
| | | consumption structure ($x_6$) | Total retail sales of social consumer goods as a percentage of GDP (%) | positive indicator |
| | | financial structure ($x_7$) | Year-end loan balance of financial institutions as a percentage of GDP (%) | positive indicator |
| | | urban-rural dichotomy ($x_8$) | Population urbanization rate (%) | positive indicator |
| | | | Per capita income ratio between urban and rural areas (%) | negative indicator |
| | Economic Green Type Development (GRD) | resource utilization ($x_9$) | Energy consumption per unit of GDP (tons of standard coal / million yuan) | negative indicator |
| | | | Electricity consumption per unit of GDP (kWh / million yuan) | negative indicator |
| | | environmental quality ($x_{10}$) | Respirable particulate matter (PM10) (μg/m3) | negative indicator |
| | | | Sulfur dioxide (SO2) ($μg/m^3$) | negative indicator |
| | | | Industrial wastewater discharge (million tons) | negative indicator |
| | Open Economy Development (OPD) | degree of foreign capital utilization ($x_{11}$) | Amount of foreign capital utilized as a proportion of GDP (%) | positive indicator |
| | | import and export scale ($x_{12}$) | Total import and export trade as a percentage of GDP (%) | positive indicator |
| | Shared Economic Development (SHD) | benefit changes ($x_{13}$) | GDP per capita (yuan/person) | positive indicator |
| | | | Average wage of employees (yuan) | positive indicator |
| | | | Number of full-time teachers per 10,000 people in general primary and secondary schools | positive indicator |
| | | | Number of health technicians per 1,000 people | positive indicator |
| | | infrastructure ($x_{14}$) | Number of passenger cars per 10,000 people | positive indicator |
| | | | Urban road area per capita ($m^2$/person) | positive indicator |
| | | | Park green space per capita ($m^2$/person) | positive indicator |

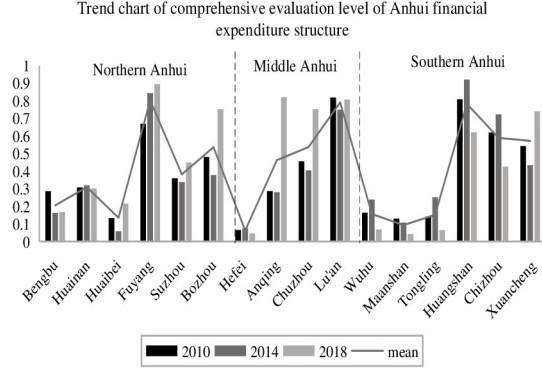

**Fig 2. Overview of fiscal expenditure structure in Anhui.**

and consumption expenditure (CE), and plots its time trend, as shown in Fig 2 (left). Meanwhile, the comprehensive evaluation level of fiscal expenditure structure (FESI) of 16 cities in Anhui in 2010, 2014, and 2018 was measured, and its change trend was plotted, as shown in Fig 2 (right). Regarding the tax structure, this paper analyzes the tax structure profile of Anhui mainly in terms of income tax (INT), commodity tax (COT), and property and behavior purpose tax (PCT), and takes other taxes (Other) into account to plot their trends, as shown in Fig 3 (left). At the same time, the data of the comprehensive evaluation level of tax structure (TSEI) of 16 cities in Anhui in 2010, 2014, and 2018 are measured by selecting the above and plotting their trends, as shown in Fig 3 (right).

(1) Overview of Anhui's financial expenditure structure.

According to Fig 2 (left), Anhui's investment expenditure (IE) shows a rising and then declining trend, rising from 15.2% in 2010 to 17.6% in 2013 and then declining to 13.0% in 2018; social expenditure (SE) as a whole shows a rising trend, with its share reaching over 80% in the past two years; the share of consumption expenditure (CE) shows a declining trend year by year, from 17.1% in 2010 to 6.4% in 2018. Among these three types of fiscal expenditures, social expenditures (SE) in Anhui rank first and are much higher than the other two types of expenditures, while investment expenditures overtook consumer expenditures in 2011 and

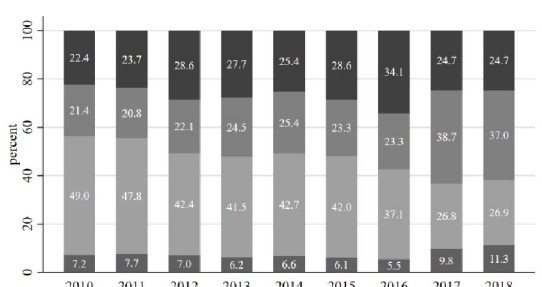
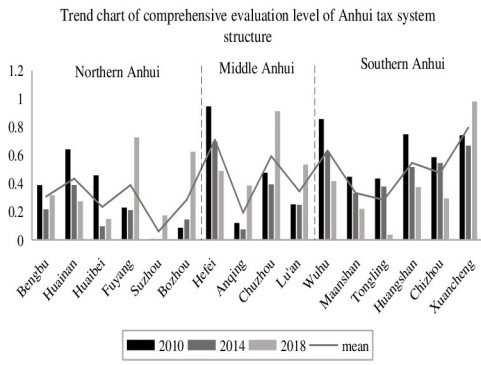

**Fig 3. Overview of Anhui tax structure.** Note: In the chart of Anhui tax structure, Other represents the share of other taxes, which is obtained by subtracting the sum of income tax, commodity tax (also known as turnover tax), property tax, and tax for behavioral purposes from Anhui local fiscal revenue; regarding commodity tax (INT), it no longer contains business tax revenue after May 2016, when the "camp reform and increase" was fully promoted.

later and ranked second, while consumer expenditures (CE) ranked last, thus it can be seen that the overall structure of fiscal expenditures in Anhui shows the characteristics of "service-oriented and investment-oriented increase". Therefore, it can be seen that the overall structure of Anhui's fiscal expenditure has the characteristics of "service-oriented and investment-oriented.

According to Fig 2 (right), there are significant differences in the comprehensive evaluation levels of fiscal expenditure structure among 16 cities in Anhui. In terms of time, Bengbu, Huabei, Suizhou, Bozhou, Anqing, Chuzhou, Liuan, and Xuancheng show a "V"-shaped structure in their comprehensive evaluation of fiscal expenditure structure, first falling and then rising; Huainan, Hefei, Wuhu, Tongling, Huangshan and Chizhou show an "inverted V"-shaped structure in their comprehensive evaluation of fiscal expenditure structure, first rising and then falling; while Fuyang shows a gradual upward trend in its comprehensive evaluation of fiscal expenditure structure, and Maanshan shows a gradual downward trend in its comprehensive evaluation of fiscal expenditure structure. Spatially, the overall mean value of the comprehensive evaluation level of fiscal expenditure structure in Fuyang, Bozhou, Chuzhou, Lu'an, Huangshan, Chizhou, and Xuancheng is at a high level, above 0.5, while the overall mean value of the comprehensive evaluation level of fiscal expenditure structure in Bengbu, Huaibei, Hefei, Wuhu, Maanshan, and Tongling is at a low level, below 0.3.

Therefore, it can be seen that the comprehensive evaluation level of fiscal expenditure structure of Anhui cities is contrary to "Wagner Principle", which may be because the fiscal system has been transformed to apply to the current market economy system, and the government has gradually withdrawn from its "overstepping" area by decentralizing its power to enterprises. In addition, there are also differences between cities in the growth rate of fiscal expenditure and the growth rate of regional GDP, for example, the growth rate of fiscal expenditure may be slower than the growth rate of GDP in cities such as Hefei and Wuhu, where economic development is more rapid.

(2) Overview of Anhui tax system structure

According to Fig 3 (left), income tax (INT), commodity tax (COT), and property and conduct purpose tax (PCT) combined account for a larger share of Anhui's fiscal revenue, reaching more than 70% in all years except 2016. Anhui Province's income tax and property and behavior tax ratios are on the rise, with the income tax ratio rising from 7.2% in 2010 to 11.3% in 2018, and the property and behavior purpose tax ratio basically fluctuating above and below the 22% level before 2016, and increasing significantly after 2017; while the commodity tax ratio shows a decreasing trend, especially after the full implementation of May 2016 After the full implementation of the "camp conversion" policy, the tax share of the property and conduct purpose exceeded the tax share of commodity tax in 2017 and 2018.

According to Fig 3 (right), there are significant differences in the comprehensive evaluation levels of the tax structure of the 16 cities in Anhui. In terms of time, Bengbu, Huaibei, Fuyang, Suizhou, Anqing, Chuzhou, Liuan, and Xuancheng show a "V" shape structure of comprehensive evaluation of tax structure, declining and then rising; Bozhou shows an increasing trend; Huainan, Hefei, Wuhu, Maanshan, Tongling, Huangshan and Chizhou show a decreasing trend of the comprehensive evaluation of tax structure. Spatially, the overall average of the comprehensive evaluation of tax structure in Hefei, Chuzhou, Wuhu, Huangshan, and Xuancheng cities is at a high level, above 0.5; the overall comprehensive evaluation of tax structure in Huaibei, Suizhou, Bozhou, Anqing, and Tongling cities is at a low level, below 0.3, so it can be seen that cities like Hefei, Chuzhou, and Wuhu are more economically developed and have higher fiscal revenues. Therefore, it can be seen that cities such as Hefei, Chuzhou, and Wuhu have more developed economies and higher fiscal revenues, and of course, the macro tax burden is also relatively high.

## 4.2. Overall situation of high-quality economic development in Anhui

The average values of Anhui's 16 prefecture-level cities are calculated based on the indicators of Anhui's comprehensive evaluation level of high-quality economic development (HQED), innovative economic development (IND), coordinated economic development (COD), green economic development (GRD), open economic development (OPD) and shared economic development (SHD), as shown in Fig 4 (left). Meanwhile, the data of the comprehensive evaluation level of high-quality economic development (HQED) of 16 cities in Anhui in 2010, 2014, and 2018 were selected to plot their change trends, as shown in Fig 4 (right).

According to Fig 4 (left), the level of high-quality economic development in Anhui shows a steady upward trend, but is still at a low level; the level of innovative economic development in Anhui shows an upward trend overall, the level of coordinated economic development and the level of open economic development shows a wave-like upward trend, the level of green economic development shows a downward and then upward trend, gradually declining since 2010 and reaching its lowest point in 2015. It reached the lowest point and then gradually rose, and the economic shared development level as a whole showed a decreasing trend. Among the five major development levels in Anhui, the economic green development level is the highest, followed by the economic coordinated development level, the economic shared development level, and the economic open development level, and the economic innovative development level is at the bottom.

According to Fig 4 (right), there are significant differences in the comprehensive evaluation levels of economic quality development among 16 cities in Anhui. In terms of time, Bengbu, Huainan, Suizhou, and Huangshan show an "inverted V" structure in their comprehensive evaluation levels of economic quality development, first declining and then rebounding; Huaibei, Wuhu, Maanshan, Tongling, and Chizhou show a declining trend; Hefei shows a "V" structure; Fuyang, Anqing, Chuzhou, Lu'an, and Xuancheng show an increasing trend. Spatially, the top five cities in terms of average degree value of comprehensive evaluation of high-quality economic development are Hefei, Wuhu, Maanshan, Tongling, and Chuzhou, of which the average value of the degree of high-quality economic development in Hefei is much higher than other cities, reaching more than 0.7, while the last two are Bozhou and Suzhou, which are located below 0.2.

## 4.3. Analysis of the coordination degree between financial and tax structure and high-quality economic development

(1) Analysis of the coordination degree between fiscal structure and high-quality economic development in Anhui.

The coordinated development degree of fiscal structure and high-quality development of Anhui economy, the calculation results are shown in Tables 4 and 5. According to the results

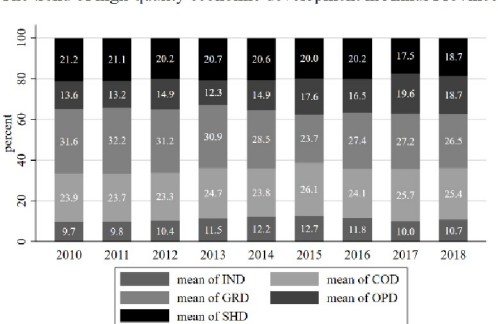

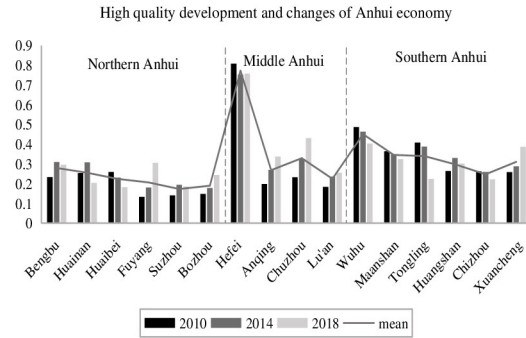

**Fig 4. Overview of high-quality economic development in Anhui.**

**Table 4. The degree of coordination between fiscal expenditure structure and high-quality economic development in Anhui Province.**

| | City Name | 2010 | 2011 | 2012 | 2013 | 2014 | 2015 | 2016 | 2017 | 2018 | Mean |
|---|---|---|---|---|---|---|---|---|---|---|---|
| Northern Anhui | Huainan | 0.525 | 0.508 | 0.606 | 0.594 | 0.560 | 0.352 | 0.524 | 0.474 | 0.484 | 0.514 |
| | Bozhou | 0.403 | 0.429 | 0.441 | 0.465 | 0.459 | 0.503 | 0.476 | 0.495 | 0.522 | 0.466 |
| | Suzhou | 0.403 | 0.442 | 0.440 | 0.434 | 0.478 | 0.469 | 0.447 | 0.469 | 0.465 | 0.450 |
| | Bengbu | 0.504 | 0.522 | 0.427 | 0.462 | 0.439 | 0.393 | 0.424 | 0.369 | 0.445 | 0.443 |
| | Fuyang | 0.351 | 0.384 | 0.358 | 0.362 | 0.415 | 0.490 | 0.476 | 0.484 | 0.588 | 0.434 |
| | Huaibei | 0.396 | 0.387 | 0.382 | 0.098 | 0.243 | 0.359 | 0.480 | 0.357 | 0.443 | 0.350 |
| | Mean | 0.430 | 0.445 | 0.442 | 0.403 | 0.432 | 0.428 | 0.471 | 0.442 | 0.491 | 0.443 |
| Middle Anhui | Chuzhou | 0.525 | 0.570 | 0.598 | 0.574 | 0.597 | 0.559 | 0.586 | 0.713 | 0.712 | 0.604 |
| | Anqing | 0.476 | 0.477 | 0.426 | 0.514 | 0.524 | 0.558 | 0.539 | 0.597 | 0.629 | 0.527 |
| | Lu'an | 0.424 | 0.448 | 0.466 | 0.505 | 0.511 | 0.537 | 0.485 | 0.520 | 0.532 | 0.492 |
| | Hefei | 0.183 | 0.183 | 0.132 | 0.161 | 0.215 | 0.354 | 0.188 | 0.087 | 0.136 | 0.182 |
| | Mean | 0.402 | 0.419 | 0.405 | 0.438 | 0.462 | 0.502 | 0.450 | 0.479 | 0.502 | 0.451 |
| Southern Anhui | Huangshan | 0.544 | 0.543 | 0.568 | 0.594 | 0.614 | 0.667 | 0.632 | 0.603 | 0.597 | 0.596 |
| | Xuancheng | 0.553 | 0.547 | 0.559 | 0.565 | 0.576 | 0.582 | 0.590 | 0.677 | 0.677 | 0.592 |
| | Chizhou | 0.555 | 0.539 | 0.548 | 0.509 | 0.544 | 0.570 | 0.547 | 0.492 | 0.512 | 0.535 |
| | Wuhu | 0.430 | 0.547 | 0.658 | 0.532 | 0.532 | 0.654 | 0.412 | 0.236 | 0.240 | 0.471 |
| | Tongling | 0.400 | 0.315 | 0.503 | 0.462 | 0.539 | 0.359 | 0.441 | 0.092 | 0.263 | 0.375 |
| | Maanshan | 0.386 | 0.229 | 0.495 | 0.380 | 0.344 | 0.257 | 0.372 | 0.181 | 0.175 | 0.313 |
| | Mean | 0.478 | 0.453 | 0.555 | 0.507 | 0.525 | 0.515 | 0.499 | 0.380 | 0.411 | 0.480 |
| | Overall mean value | 0.441 | 0.442 | 0.476 | 0.451 | 0.474 | 0.479 | 0.476 | 0.428 | 0.464 | 0.459 |

**Table 5. The degree of coordination between tax structure and high-quality economic development in Anhui Province.**

| | City Name | 2010 | 2011 | 2012 | 2013 | 2014 | 2015 | 2016 | 2017 | 2018 | Mean |
|---|---|---|---|---|---|---|---|---|---|---|---|
| Northern Anhui | Huainan | 0.543 | 0.528 | 0.637 | 0.628 | 0.582 | 0.534 | 0.543 | 0.472 | 0.478 | 0.550 |
| | Bozhou | 0.523 | 0.548 | 0.554 | 0.543 | 0.497 | 0.550 | 0.563 | 0.555 | 0.553 | 0.543 |
| | Suzhou | 0.396 | 0.405 | 0.403 | 0.416 | 0.439 | 0.503 | 0.486 | 0.509 | 0.599 | 0.462 |
| | Bengbu | 0.553 | 0.538 | 0.575 | 0.524 | 0.335 | 0.306 | 0.324 | 0.289 | 0.402 | 0.427 |
| | Fuyang | 0.316 | 0.320 | 0.350 | 0.444 | 0.397 | 0.490 | 0.476 | 0.504 | 0.533 | 0.425 |
| | Huaibei | 0.031 | 0.051 | 0.085 | 0.156 | 0.011 | 0.265 | 0.175 | 0.413 | 0.422 | 0.179 |
| | Mean | 0.394 | 0.398 | 0.434 | 0.452 | 0.377 | 0.441 | 0.428 | 0.457 | 0.498 | 0.431 |
| Middle Anhui | Chuzhou | 0.930 | 0.877 | 0.798 | 0.814 | 0.850 | 0.874 | 0.891 | 0.769 | 0.752 | 0.839 |
| | Anqing | 0.525 | 0.569 | 0.612 | 0.609 | 0.594 | 0.615 | 0.623 | 0.714 | 0.714 | 0.619 |
| | Lu'an | 0.455 | 0.467 | 0.460 | 0.471 | 0.491 | 0.555 | 0.518 | 0.538 | 0.549 | 0.500 |
| | Hefei | 0.374 | 0.253 | 0.320 | 0.372 | 0.280 | 0.431 | 0.450 | 0.585 | 0.598 | 0.407 |
| | Mean | 0.571 | 0.542 | 0.548 | 0.566 | 0.554 | 0.619 | 0.620 | 0.651 | 0.653 | 0.592 |
| Southern Anhui | Huangshan | 0.757 | 0.733 | 0.723 | 0.730 | 0.721 | 0.732 | 0.721 | 0.638 | 0.640 | 0.711 |
| | Xuancheng | 0.542 | 0.543 | 0.565 | 0.576 | 0.581 | 0.608 | 0.605 | 0.676 | 0.671 | 0.596 |
| | Chizhou | 0.549 | 0.559 | 0.587 | 0.602 | 0.618 | 0.647 | 0.613 | 0.577 | 0.574 | 0.592 |
| | Wuhu | 0.630 | 0.578 | 0.556 | 0.587 | 0.582 | 0.599 | 0.582 | 0.511 | 0.502 | 0.570 |
| | Tongling | 0.556 | 0.536 | 0.532 | 0.502 | 0.554 | 0.583 | 0.528 | 0.485 | 0.498 | 0.530 |
| | Maanshan | 0.648 | 0.605 | 0.625 | 0.602 | 0.619 | 0.412 | 0.365 | 0.171 | 0.175 | 0.469 |
| | Mean | 0.614 | 0.592 | 0.598 | 0.600 | 0.613 | 0.597 | 0.569 | 0.510 | 0.510 | 0.578 |
| | Overall mean value | 0.520 | 0.507 | 0.524 | 0.536 | 0.509 | 0.544 | 0.529 | 0.525 | 0.541 | 0.526 |

in Tables 4 and 5, it can be seen that the mean value of coordination between fiscal expenditure structure and high-quality economic development in Anhui Province fluctuates in the range of 0.42 to 0.48. According to the coordination criteria in Table 1, the level of coordination between fiscal expenditure structure and high-quality economic development in Anhui Province is "nearly disorder". The coordination degree of tax structure and high-quality economic development in Anhui Province fluctuates between 0.50 and 0.55, and the overall spiral is increasing. The tax system structure is in a state of "barely coordination" with high-quality economic development. In summary, the level of coordination between the fiscal structure and high-quality economic development in Anhui Province is still low, and the overall situation is "nearly disorder" or "barely coordination".

(2) Analysis of regional heterogeneity in coordination.

The Huai River and the Yangtze River are used as the boundary to divide Anhui Province into three regions: northern, middle, and southern Anhui. (The northern Anhui region includes six cities of Suizhou, Huabei, Bengbu, Fuyang, Huainan, and Bozhou; the middle Anhui region includes four cities of Hefei, Lu'an, Chuzhou, and Anqing; the southern Anhui region includes six cities of Huangshan, Wuhu, Maanshan, Tongling, Xuancheng, and Chizhou.) According to Tables 4 and 5, the coordination degree of financial expenditure structure and economic quality development in northern Anhui Province fluctuates between 0.4 and 0.5 and shows an overall rising trend, but the rise is not large, and it is basically in the state of "nearly disorder".

For the middle Anhui region, except for 2012, the coordination degree of fiscal expenditure structure and economic quality development has been increasing from 2010 to 2015, from 0.402 to 0.502, an increase of 24.9%, and the level of coordinated development has been greatly improved. After 2016, the coordination between fiscal expenditure structure and economic quality development has shown a new round of improvement, breaking through 0.5 and reaching the "barely coordination" status. The coordination degree of tax structure and high-quality economic development in middle Anhui region fluctuated between 0.5 and 0.59 from 2010 to 2014, with a trend of slight decrease, and crossed 0.6 after 2015, from 0.619 in 2015 to 0.653 in 2018, with an increase of 5.2%, from "barely coordinated" to "primary coordinated" status. This may be attributed to the full implementation of the "VAT reform" policy on May 1, 2016, which further optimized the tax structure and enhanced the synergy with high-quality economic development.

For the southern Anhui region, the coordination degree of fiscal expenditure structure and economic quality development from 2010 to 2018 fluctuates widely and has no obvious pattern, for example, the coordination degree in 2017 is 0.38, which is in the state of "mild dissonance", and the coordination degrees in 2010, 2011, 2016 and 2018 are in the state of "on the verge of dissonance". The coordination degree of fiscal expenditure structure and economic quality development in the rest of the years is in the state of "barely coordinated". Except for 2010, the coordination degree of tax structure and economic quality development in Southern Anhui region continued to increase from 2011 to 2014, from 0.592 to 0.613, and after 2015, it showed a decreasing trend, from 0.597 to 0.510, i.e. the coordination degree of tax structure and economic quality development in South Anhui region showed a rising and then decreasing trend. The possible reason is that the southern Anhui region has not grasped the opportunity of "VAT reform" to optimize the tax structure and bring into play the synergy between the tax structure and high-quality economic development, thus regressing from the state of "primary coordination" to the state of "barely coordination".

In terms of spatial differences in the coordination of fiscal structure and high-quality economic development in the three major regions, in terms of the coordination of fiscal

expenditure structure and high-quality economic development, the coordination of Southern Anhui region was higher than that of Northern Anhui region from 2010 to 2013, and Middle Anhui region ranked last; from 2014 to 2016, the coordination of Middle Anhui region surpassed that of Northern Anhui region, and the ranking changed to Southern Anhui region, Middle Anhui region and Northern Anhui region; in 2017 and 2018, the coordination of Middle Anhui region and Northern Anhui region surpassed that of Southern Anhui region, and the ranking changed to Middle Anhui region, Northern Anhui region and South Anhui region. From the coordination degree of tax structure and high-quality economic development, the coordination degree of Southern Anhui region was higher than that of Middle Anhui region from 2010 to 2014, and Northern Anhui region ranked last and far behind the other two regions, after 2015, Middle Anhui region far surpassed Southern Anhui region, and the ranking changed to Middle Anhui region, Southern Anhui region and Northern Anhui region, and Northern Anhui region also had the momentum to surpass Southern Anhui region.

In summary, the overall coordination of fiscal expenditure structure, taxation system, and high-quality economic development in Southern Anhui shows a decreasing trend, while the overall coordination of Middle Anhui and Northern Anhui shows an increasing trend, so that Southern Anhui has been or will be surpassed by Northern and Middle Anhui, and the growth rate of Middle Anhui is faster than Northern Anhui.

(3) Analysis of coordination differences between cities.

According to Tables 4, 5 and Fig 5, from the average value of coordination between fiscal expenditure structure and economic quality development from 2010 to 2018 (Fig 5 left), the cities with the average value of coordination above 0.5 are Chuzhou, Huangshan, Xuancheng, Chizhou, Anqing, and Huainan, of which Chuzhou has an average value of coordination above 0.6, reaching the "primary coordination "state, while the other five cities in the 0.5 or more, in the "barely coordinated" state. The average value of coordination of Lu'an, Wuhu, Bozhou, Suizhou, Bengbu, and Fuyang is between 0.4 and 0.5, which is in the state of "nearly disorder"; while the last cities are Tongling, Huaibei, Maanshan, and Hefei, whose average value of coordination is below 0.4, especially the average value of coordination of Hefei is 0.182, which is at the bottom of Anhui cities. Hefei is at the bottom of the list, in a state of "serious disorder", while the other three cities are in a state of "mild disorder".

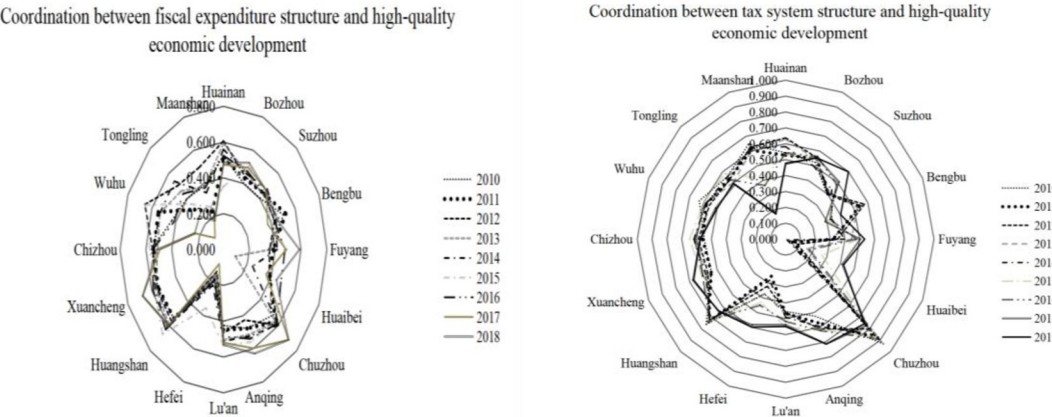

**Fig 5. Coordination between fiscal structure and high-quality economic development in Anhui cities.**

From the average value of coordination between tax structure and high-quality economic development in 2010–2018 (Fig 5 right), the top three cities are Hefei, Wuhu, and Chuzhou, which are at the stage of "good coordination", "intermediate coordination" and "primary coordination" respectively, with a high level of coordinated development; followed by Xuancheng, Huangshan, Maanshan, Huainan, Bengbu, Chizhou and Liuan, whose coordinated development levels of tax structure and high-quality economic development are in the state of "barely coordinated". However, Tongling, Fuyang, Huaibei, Bozhou, and Anqing are all in a state of "nearly disorder" in terms of the coordination between their tax structure and high-quality economic development, while the average value of coordination in Suzhou is far behind other cities in Anhui, ranking at the bottom and in a state of "serious disorder".

## 5. Conclusion and policy implications

This paper selects the panel data of 16 cities in Anhui from 2010–2018 as samples, measures the index system of fiscal structure and high-quality economic development in Anhui based on the entropy weight method, and introduces the coupled coordination degree model to empirically analyze the coordinated development level of fiscal structure and high-quality economic development in Anhui. The results show that

1. Anhui's fiscal structure has different characteristics in time and space. As for the structure of fiscal expenditure, the analysis of the proportion of investment expenditure, social expenditure, and consumption expenditure in Anhui shows that the overall structure is "service-oriented and investment-oriented". By comparing the comprehensive evaluation level of fiscal expenditure structure of Anhui cities, there is a phenomenon that contradicts the "Wagner's Law" on the whole; from the viewpoint of tax structure, the proportion of income tax in Anhui is on the rise. The overall trend of commodity tax is decreasing, and the proportion of property and behavior tax is increasing in general, and exceeds the proportion of commodity tax in 2017 and 2018. By comparing the comprehensive evaluation level of tax structure of each city in Anhui, cities like Huabei, Suizhou, and Bozhou are at a lower level, while cities like Hefei, Chuzhou, and Wuhu, with more developed economies and higher fiscal revenues, are at a higher level overall.

2. Anhui's economic high-quality development level shows a steady upward trend, but is still at a low level. Among the five dimensions of economic high-quality development in Anhui, the highest level of economic green development is followed by the level of economic coordinated development, the level of economic shared development, and the level of economic open development, and in the last place is the level of economic innovative development. Spatially, the top five cities in terms of the average value of comprehensive evaluation level of economic high-quality development are Hefei, Wuhu, Maanshan, Tongling, and Chuzhou, among which the average value of comprehensive evaluation level of economic high-quality development in Hefei is much higher than other cities, reaching more than 0.7, while the last two are Bozhou and Suizhou, which are located below 0.2.

3. The level of coordination between fiscal structure and high-quality economic development is still low, and the overall situation is "nearly disorder" or "barely coordinated"; for the three major regions, the overall coordination of fiscal expenditure structure, tax system and high-quality economic development in southern Anhui shows a decreasing trend, while the overall coordination in middle and northern Anhui shows an increasing trend, so that southern Anhui has been or will be surpassed by northern and middle Anhui, and the growth rate of middle Anhui is faster than northern Anhui.

4. From the perspective of Anhui cities, the top five cities in terms of the average value of coordination between fiscal expenditure structure and high-quality economic development are Chuzhou, Huangshan, Xuancheng, Chizhou, and Anqing, which are in the state of "primary coordination" or "barely coordination", while the last five cities are Tongling, Huaibei, Maanshan, and Hefei, and Hefei is in a state of "serious disorder". The top three cities in terms of the average value of coordination between tax structure and high-quality economic development are Hefei, Wuhu, and Chuzhou, which are in the stage of "good coordination", "intermediate coordination" and "primary coordination" respectively, with a high level of coordinated development; the last cities are Tongling, Fuyang, Huabei, Bozhou, and Anqing, which are in the state of "nearly disorder", while the average value of coordination in Suizhou is far behind other cities in Anhui, ranking at the bottom and in the state of "serious disorder".

Based on the above research findings, this paper puts forward the following policy recommendations:

1. Optimize the fiscal structure and drive high-quality economic development. Firstly, the scale of fiscal expenditure should be increased moderately, the structure of fiscal expenditure should be optimized, and the proportion of investment expenditure, social expenditure, and consumer expenditure should be reasonably allocated. Increase social expenditures such as education expenditures, science and technology expenditures, medical health and family planning expenditures, and energy conservation and environmental protection expenditures, as well as investment expenditures such as general public service expenditures and transportation expenditures with targets and priorities, to give full play to the positive driving effect of high-quality economic development. Second, moderate control of the tax burden, optimize the tax structure, such as sub-regional differences in the appropriate reduction of VAT and other business tax levies, to give enterprises R & D subsidies, tax incentives, and other policies to promote the innovative development of Anhui's economy to drive high-quality economic development.

2. For cities such as Tongling, Huaibei, Maanshan, and Hefei, the level of coordinated development between fiscal expenditure structure and high-quality economic development is low and still in a state of dysfunction. Fiscal expenditure structure should be continuously optimized to drive the level of high-quality economic development to continuously improve and support each other to realize a virtuous cycle of spiral upward trend and finally realize high-quality coordinated development. For cities such as Tongling, Fuyang, Huaibei, Bozhou, Anqing, and Suzhou, the level of coordination between the tax structure and high-quality economic development is low. Measures should be taken to optimize the tax structure, boost high-quality economic development, continuously increase fiscal revenue, and achieve a new round of optimization of the tax structure.

The indicator system of China's high-quality economic development is constructed from five aspects: innovation, coordination, green, openness, and sharing, and the development level coming out of this construction will be changed by the number of secondary indicators. It is not strictly verified whether the measured values really represent the level of China's high quality economic development.

## Acknowledgments

The authors are very grateful to the reviewers for their valuable comments and the editors for their hard work.

## Author Contributions

**Conceptualization:** Yunlei Zhou, Xiaoyu He.

**Data curation:** Yunlei Zhou, Xiaoyu He, Shengsheng Li.

**Formal analysis:** Yunlei Zhou, Shengsheng Li.

**Funding acquisition:** Yunlei Zhou, Xiaoyu He.

**Methodology:** Yunlei Zhou, Shengsheng Li.

**Visualization:** Xiaoyu He.

**Writing – original draft:** Yunlei Zhou, Xiaoyu He, Shengsheng Li.

**Writing – review & editing:** Yunlei Zhou, Xiaoyu He, Shengsheng Li.

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
