## [Decision Letter · Decision Letter 0]

16 Jan 2023

PONE-D-22-33769Research on the Coordination of Fiscal Structure and High-Quality Economic Development: An Empirical Analysis with the Example of Anhui Province in ChinaPLOS ONE

Dear Dr. Li,

Thank you for submitting your manuscript to PLOS ONE. After careful consideration, we feel that it has merit but does not fully meet PLOS ONE’s publication criteria as it currently stands. Therefore, we invite you to submit a revised version of the manuscript that addresses the points raised during the review process.

We look forward to receiving your revised manuscript.

Kind regards,

Chenguel Mohamed Bechir, Phd HDR

Academic Editor

PLOS ONE

Journal Requirements:

"Grant numbers awarded to each author."

"For this research, the first author(Yunlei Zhou) was supported by the 2021 Research Projects on Humanities and Social Sciences in Anhui Universities under grant SK2021A0229, the Project of Anhui Ecological and Economic Development Research Center of China under grant AHST2019016, AHST2021010, AHST2019011 and AHST2021007, and the Social Science Planning in Anhui Province of China under grant AHSKZ2019D018. The second author(Xiaoyu He) was supported the by the National Social Science Youth Fund of China under grant 19CJY055."

"Grant numbers awarded to each author."

7. Please upload a new copy of Figure 5 as the detail is not clear. Please follow the link for more information:

https://blogs.plos.org/plos/2019/06/looking-good-tips-for-creating-your-plos-figures-graphics/

https://blogs.plos.org/plos/2019/06/looking-good-tips-for-creating-your-plos-figures-graphics/

Reviewers' comments:

Reviewer's Responses to Questions

**Comments to the Author**

1. Is the manuscript technically sound, and do the data support the conclusions?

Reviewer #1: Yes

Reviewer #2: Yes

2. Has the statistical analysis been performed appropriately and rigorously? 

Reviewer #1: Yes

Reviewer #2: Yes

3. Have the authors made all data underlying the findings in their manuscript fully available?

Reviewer #1: Yes

Reviewer #2: Yes

4. Is the manuscript presented in an intelligible fashion and written in standard English?

Reviewer #1: No

Reviewer #2: Yes

5. Review Comments to the Author

Reviewer #1: The article is very interesting but some improvements are needed:

-Add recent references

-Correct spelling and grammar mistakes

-Add a literature review to the article

-Add a conclusion to the article

Reviewer #2: -This research paper discusses an interesting problem of "measuring the index system of the fiscal structure and high-quality economic development in Anhui using the entropy weight method and and empirically analyzes the coordinated development

level of fiscal structure and high-quality economic development in Anhui using coupled coordination degree model". The literature review is modest, the authors will need to analyse more recent research related to the second,Fiscal structure and coordinated economic development , the third,Fiscal structure and green economic development, the fourth,Fiscal structure and open economic development and the fith, areas (pages 10,11,12)

-the authors should document the strategic, tactical and operational implications

6. PLOS authors have the option to publish the peer review history of their article (what does this mean?). If published, this will include your full peer review and any attached files.

Reviewer #1: No

Reviewer #2: No

---

## [Author Response · Author response to Decision Letter 0]

26 Feb 2023

Dear professor, 

We are grateful to the reviewers for finding the paper of some value and providing valuable comments. We have revised the paper in accordance with the reviewers' comments.

Reviewer #1: The article is very interesting but some improvements are needed:

-Add recent references

-Correct spelling and grammar mistakes

-Add a literature review to the article

-Add a conclusion to the article

Response: 

Thank you very much for the suggestions you gave us, we have revised the paper as follows, adding recent references as well as a literature review. We have rechecked the grammar and spelling and corrected any errors. Also, the fifth part of our paper is the conclusion and policy recommendations.

All changes in our paper are marked in red in the Revised Manuscript with Track Changes.docx file.

Reviewer #2: -This research paper discusses an interesting problem of "measuring the index system of the fiscal structure and high-quality economic development in Anhui using the entropy weight method and and empirically analyzes the coordinated development

level of fiscal structure and high-quality economic development in Anhui using coupled coordination degree model". The literature review is modest, the authors will need to analyse more recent research related to the second,Fiscal structure and coordinated economic development , the third,Fiscal structure and green economic development, the fourth,Fiscal structure and open economic development and the fith, areas (pages 10,11,12)

Response: 

Thank you very much for your approval of our paper and for your valuable comments, we have added the latest literature on the five areas you provided. And the additions are marked in red.

---

## [Decision Letter · Decision Letter 1]

19 Apr 2023

PONE-D-22-33769R1Research on the Coordination of Fiscal Structure and High-Quality Economic Development: An Empirical Analysis with the Example of Anhui Province in ChinaPLOS ONE

Dear Dr. Li,

Thank you for submitting your manuscript to PLOS ONE. After careful consideration, we feel that it has merit but does not fully meet PLOS ONE’s publication criteria as it currently stands. Therefore, we invite you to submit a revised version of the manuscript that addresses the points raised during the review process.

We look forward to receiving your revised manuscript.

Kind regards,

Chenguel Mohamed Bechir, Phd HDR

Academic Editor

PLOS ONE

Journal Requirements:

Additional Editor Comments (if provided):

- please Correct spelling and grammar mistakes

- please Add recent references

-please Add a future implication in conclusion 

The literature review is modest, you will need to analyse more recent research related to the second,Fiscal structure and coordinated economic development , the third,Fiscal structure and green economic development, the fourth,Fiscal structure and open economic development and the fith, areas (pages 10,11,12).

-the authors should clarify  the strategic and operational implications

thank you for making all corrections

Reviewers' comments:

Reviewer's Responses to Questions

**Comments to the Author**

1. If the authors have adequately addressed your comments raised in a previous round of review and you feel that this manuscript is now acceptable for publication, you may indicate that here to bypass the “Comments to the Author” section, enter your conflict of interest statement in the “Confidential to Editor” section, and submit your "Accept" recommendation.

Reviewer #2: All comments have been addressed

Reviewer #3: All comments have been addressed

2. Is the manuscript technically sound, and do the data support the conclusions?

Reviewer #2: Yes

Reviewer #3: Yes

3. Has the statistical analysis been performed appropriately and rigorously? 

Reviewer #2: Yes

Reviewer #3: Yes

4. Have the authors made all data underlying the findings in their manuscript fully available?

Reviewer #2: Yes

Reviewer #3: Yes

5. Is the manuscript presented in an intelligible fashion and written in standard English?

Reviewer #2: Yes

Reviewer #3: No

6. Review Comments to the Author

Reviewer #2: the autors have made the requested corrections: he has corrected the litterarure review and added the implications

Reviewer #3: * The authors study this problematic: Research on the Coordination of Fiscal Structure and High-Quality Economic

Development: An Empirical Analysis with the Example of Anhui Province in China

* I consider that the problem is original and recent.

* the article usually contains stylized facts. it is not an empirical contribution or a modeling.

* For the conclusion, it is necessary to interpret the results well and to give other

economic, financial and political implications.

* Usually, the research is appropriately motivated.

* Accept for publication

7. PLOS authors have the option to publish the peer review history of their article (what does this mean?). If published, this will include your full peer review and any attached files.

Reviewer #2: No

Reviewer #3: No

---

## [Author Response · Author response to Decision Letter 1]

19 May 2023

Dear editor and reviewers, 

We are very grateful for your valuable comments. Our modifications are described as follows:

Editor:

Response: 

Our references are in compliance with the criteria, where reference [31] is in Table 3. Because Table 3 is too large, it is not in the manuscript but is uploaded as an attachment.

2. - please Correct spelling and grammar mistakes

- please Add recent references

- The literature review is modest, you will need to analyse more recent research related to the second,Fiscal structure and coordinated economic development , the third,Fiscal structure and green economic development, the fourth,Fiscal structure and open economic development and the fith, areas (pages 10,11,12).

Response: 

Thank you very much Dr. Bechir for raising these detailed questions. We are very sorry for not formally responding to your comments in the first revision, which were duplicated by your comments and those of the reviewers, and we have made changes. All changes in our paper are marked in red in the Revised Manuscript with Track Changes.docx file. 

We have made changes to grammar and other issues.

We conducted a review of the relevant literature in the introduction, citing 27 recent papers. The main body of our paper is an analysis of the coherence between fiscal structure and quality economic development. We have constructed the indicators of high-quality economic development based on the characteristics of China's economy in five aspects: innovation, coordination, green, openness and sharing. Most of the papers focus on the general analysis, and we are further thinking in five areas. As a result, there is relatively little literature in this area.

3. - the authors should clarify the strategic and operational implications

The fifth part of our manuscript is the conclusion and policy recommendations, which are based on the empirical findings. And we have added a reflection on the limitations of the paper.

Reviewer #2: the autors have made the requested corrections: he has corrected the litterarure review and added the implications

Reviewer #3: * The authors study this problematic: Research on the Coordination of Fiscal Structure and High-Quality Economic Development: An Empirical Analysis with the Example of Anhui Province in China.

* I consider that the problem is original and recent.

Response: 

Thank you again for your previous comments. And thank you for your approval of the revised manuscript.

---

## [Editor Report · Decision Letter 2]

1 Jun 2023

Research on the Coordination of Fiscal Structure and High-Quality Economic Development: An Empirical Analysis with the Example of Anhui Province in China

PONE-D-22-33769R2

Dear Dr. Shengsheng Li

We’re pleased to inform you that your manuscript has been judged scientifically suitable for publication and will be formally accepted for publication once it meets all outstanding technical requirements.

Kind regards,

Chenguel Mohamed Bechir, Phd HDR

Academic Editor

PLOS ONE

Additional Editor Comments (optional):

thank you for your revision
---

## [Editor Report · Acceptance letter]

6 Jun 2023

PONE-D-22-33769R2 

Research on the coordination of fiscal structure and high-quality economic development: An empirical analysis with the example of Anhui province in China 

Dear Dr. Li:

I'm pleased to inform you that your manuscript has been deemed suitable for publication in PLOS ONE. Congratulations! Your manuscript is now with our production department. 

Kind regards, 

on behalf of

Dr. Chenguel Mohamed Bechir 

Academic Editor

PLOS ONE